# Association between Chlamydia and routine place for healthcare in the United States: NHANES 1999–2016

**Cornelius D. Jamison**[1,2]*, **Margaret Greenwood-Ericksen**[3], **Caroline R. Richardson**[1,2], **Hwajung Choi**[4], **Tammy Chang**[1,2,5]

**1** Department of Family Medicine, University of Michigan, Ann Arbor, MI, United States of America, **2** Institute for Healthcare Policy and Innovation, University of Michigan, Ann Arbor, MI, United States of America, **3** Department of Emergency Medicine, University of New Mexico, Albuquerque, NM, United States of America, **4** Department of Internal Medicine, University of Michigan, Ann Arbor, MI, United States of America, **5** National Clinician Scholars Program, University of Michigan, Ann Arbor, MI, United States of America

* jcorneli@med.umich.edu

## Abstract

### Background

The United States is experiencing a surge in *Chlamydia trachomatis* (CT) infections representing a critical need to improve sexually transmitted infection (STI) screening and treatment programs. To understand where patients with STIs seek healthcare, we evaluated the relationship between CT infections and the place where individuals report usually receiving healthcare.

### Methods

Our study used a nationally representative sample from the National Health and Nutrition Examination Survey (NHANES) from 1999 to 2016. The study population is adult patients, aged 18 to 39 years in whom a urine CT screen was obtained. Logistic regression models were used to determine if location of usual healthcare was predictive of a positive urine CT screen result. Models were adjusted for known confounders including age, gender, race/ethnicity, education, and insurance status.

### Results

In this nationally representative sample (n = 19,275; weighted n = 85.8 million), 1.9% of individuals had a positive urine CT result. Participants reported usually going to the doctor's office (70.3%), "no place" (24.8%), Emergency Department (ED) (3.3%), or "other" place (1.7%) for healthcare. In adjusted models, the predicted probability of having a positive urine CT result is higher (4.9% vs 3.2%, p = 0.022; OR = 1.58) among those that reported the ED as their usual place for healthcare compared to those that reported going to a doctor's office or clinic.

**Data Availability Statement:** All relevant data are from the National Health and Nutrition Examination Survey (NHANES) and publicly available from the

CDC website: https://wwwn.cdc.gov/nchs/nhanes/Default.aspx.

**Funding:** CDJ acknowledges funding from the National Clinician Scholars Program and the United States Department of Veterans Affairs supporting his work on this project. The funders had no role in study design, data collection and analysis, decision to publish, or preparation of the manuscript.

**Competing interests:** The authors have declared that no competing interests exist.

**Abbreviations:** CDC, Centers for Disease Control and Prevention; CT, *Chlamydia trachomatis*; ED, Emergency Department; NHANES, National Health and Nutrition Examination Survey; PCP, Primary Care Provider; STI, Sexually Transmitted Infection.

## Conclusions

Individuals having a positive urine CT screen are associated with using the ED as a usual source for healthcare. Understanding this association has the potential to improve STI clinical and policy interventions as the ED may be a critical site in combatting the record high rates of STIs.

## Introduction

The United States (US) currently has the highest ever recorded rate of *Chlamydia trachomatis* (CT) infection [1]. There were 1.8 million reported cases of CT in 2018—resulting in an increase of 19%, since 2014 according to the Centers for Disease Control and Prevention (CDC) [1]. Young adults aged 18–39 years accounted for an estimated 79% of these reported CT cases in 2018 [1]. CT infections result in a range of health complications including urethritis, epididymitis, and pelvic inflammatory disease that could be prevented through regular office visits and preventive care [2]. But as traditional sexually transmitted infection (STI) clinics declined, due to state and local budget constraints, individuals have been required to seek care elsewhere [3].

Several studies have identified key risk factors associated with having an STI, including high risk sexual behavior, previous history of an STI, and illicit drug abuse [4, 5]. Geographical and social factors have also been evaluated to help determine those more vulnerable to contracting an STI [6, 7]. Where individuals with STIs choose to obtain screening and treatment has not been described in the literature, yet recent studies have found an increase in the number of visits to the Emergency Department (ED) involving an STI diagnosis [1, 8]. Determining where individuals chose to obtain preventive care, including STI screening and treatment has important implications for the current epidemic.

The purpose of this study was to use a nationally representative dataset to evaluate the relationship between CT infection (measured by urine CT screen) and self-reported place most often visited for healthcare. To our knowledge, no population-level studies have evaluated this relationship. Understanding this relationship may guide future interventions to improve the healthcare sites individuals frequent to obtain STI screening and treatment.

## Methods

Our study used a nationally representative sample from the National Health and Nutrition Examination Survey (NHANES), which combines interviews and physical examinations with laboratory testing to assess the health and nutritional status of adults and children in the US [9]. The complex study design of NHANES uses a subset of the US population, representing the unweighted number of individuals. The weighting measurements used in analysis account for the US population. NHANES participants were not seeking nor being provided medical care at the time of their participation. Data from the nine most recent NHANES cohorts (1999 to 2016) were used for analyses. From the Hospital Utilization & Access to Care Questionnaire individuals were first asked, "Is there a place that you usually go when you are sick or you need advice about your health?" (variable HUQ030). Then they were asked, "What kind of place do you go to most often—a clinic, doctor's office, emergency room, or some other place?" (variable HUQ040). Beginning in 2013 the question for variable HUQ040 was modified but collected the same information (variable HUQ041). If the participants answered "Yes" to

HUQ030, then they were asked "What kind of place is it—a clinic, doctor's office, emergency room, or some other place?" If they provided any answer other than "Yes" to HUQ030, then they were asked "What kind of place do you go to most often—a clinic, doctor's office, emergency room, or some other place? 'Place of healthcare' categories were developed based on NHANES participant's responses to HUQ040 and HUQ041. For 'doctor's office/clinic' we combined clinic or health center, doctor's office or HMO, and hospital outpatient department responses. For 'ED' we used hospital emergency room, for 'other' we used some other place, and for 'no place' we combined doesn't go to one place most often, refused, and don't know responses.

Our analysis included urine CT samples (variable URXUCL) from the public data of individuals aged 18–39 years, the ages eligible for NHANES participation. CT was chosen for this study because it is the most commonly reported STI in the US and has been included in the NHANES laboratory data consecutively for multiple waves [1]. The outcome measure was a positive urine CT result. Urine samples were collected at the medical examination and were processed, stored, and shipped to the Division of AIDS, STD, and TB Laboratory Research, National Center for Infectious Diseases, National Centers for Disease Control and Prevention in Atlanta, GA for analysis [9]. Urine samples were processed through different reliable assays as the waves progressed [9]. NHANES waves 1999–2002 used the LCx *Chlamydia trachomatis* assay, 2003–08 used the BDProbeTec CT *Chlamydia trachomatis* and *Neisseria gonorrhoeae* Amplified DNA Assays, waves 2009–16 used the Gen-Probe APTIMA Combo 2 *Chlamydia trachomatis* Assay [10–12].

Bivariate analyses between urine CT screen and covariates were performed using independent sample t-tests as appropriate. Logistic regression was performed with a positive urine CT screen (urine samples) as the outcome. Models were adjusted for sociodemographic covariates including age, gender, race/ethnicity, education, and insurance status and complex sampling design. To address the potential non-linear effect of age, we included the quadratic form age squared (shown as 'Age x Age' in Table 2) rather than linear form, age alone. We performed sensitivity analyses that included family income-to-poverty ratio (in addition to education) as a surrogate for socio-economic status in our model and found no significant changes in the outcome. All analyses were performed using STATA 14.2 (StataCorp LP) and accounted for appropriate complex survey weights to allow for national inferences that represent the US non-institutionalized civilian population [13]. The University of Michigan Institutional Review Board deemed this study to be exempt from regulation.

## Results

Data collected from NHANES between 1999–2016 resulted in an unweighted sample size of 19,514 individuals (weighted N = 86.8 million) aged 18 to 39 years with complete CT urine sample data. The unweighted sample size decreased to 19,275 individuals (weighted N = 85.8 million) during multivariable analysis due to missing in covariates. Table 1 presents the sociodemographic characteristics of the sample by routine place of healthcare. In our overall weighted sample, the median age was 28.6 years, 50.4% were women, and 60.3% were non-Hispanic white. Of our overall sample, 1.9% of participating individuals had a positive urine CT result. Participants reported usually going to the "doctor's office" (70.3%), "no place" (24.8%), "emergency department" (ED) (3.3%), or "other place" (1.7%) for healthcare (numbers not shown in Tables).

Table 2 provides odds ratios from the bivariate analyses and multivariable analysis models. In bivariate analyses, a positive urine CT result was significantly associated with the ED as the type of place most often visited for healthcare (OR = 2.68, p<0.01, CI 1.18–3.95; Table 2), this

**Table 1. Estimated population of sociodemographic characteristics among 18–39 years old by routine place for healthcare, NHANES, 1999 to 2016.**

| | Overall (n = 19,514) | Doctor's Office | ED | No Place | ED vs. Doctor's Office (p) | No Place vs. Doctor's Office (p) |
|---|---|---|---|---|---|---|
| **Age (median)** | 28.6 | 29.0 | 27.8 | 27.5 | <0.001 | <0.001 |
| **CT Prevalence, %** | 1.9 | 1.6 | 4.3 | 2.2 | <0.001 | 0.010 |
| **Female, %** | 50.4 | 57.3 | 43.3 | 32.8 | <0.001 | <0.001 |
| **Race/Ethnicity, %** | | | | | | |
| Non-Hispanic White | 60.3 | 64.1 | 46.6 | 51.1 | <0.001 | <0.001 |
| Non-Hispanic Black | 12.9 | 13.0 | 29.8 | 10.6 | <0.001 | <0.001 |
| Mexican American | 12.0 | 9.5 | 8.3 | 19.6 | 0.273 | <0.001 |
| Other Hispanic | 7.2 | 6.3 | 10.2 | 9.4 | <0.001 | <0.001 |
| Other Ethnicity | 7.6 | 7.1 | 5.2 | 9.3 | 0.082 | <0.001 |
| **Education, %** | | | | | | |
| Less than 9th Grade | 4.2 | 2.9 | 7.6 | 7.4 | <0.001 | <0.001 |
| 9th-11th Grade | 14.1 | 12.4 | 24.3 | 18.1 | <0.001 | <0.001 |
| HS Graduate/GED | 23.9 | 23.0 | 34.0 | 25.4 | <0.001 | 0.006 |
| Some College/Assoc. | 33.7 | 35.1 | 26.7 | 29.9 | <0.001 | <0.001 |
| College Grad or higher | 24.1 | 26.6 | 7.4 | 19.2 | <0.001 | <0.001 |
| **Insurance Status, %** | | | | | | |
| Private | 66.2 | 74.3 | 37.4 | 46.7 | <0.001 | <0.001 |
| Medicaid | 6.3 | 7.3 | 11.1 | 3.3 | 0.005 | <0.001 |
| Uninsured | 27.5 | 18.3 | 51.6 | 50.1 | <0.001 | <0.001 |

remained significant after adjusting for the confounders in the multivariable analysis (OR = 1.58, p = 0.022, CI 1.07–2.36; Table 2). The predicted probability of having a positive urine CT result in individuals, aged 18 to 39 years, is higher (4.9% vs 3.2%, p = 0.022; OR = 1.58) among those that reported the ED as the place most often visited for healthcare compared to those that reported visiting a doctor's office or clinic. We further examined this association through a multivariable analysis model adjusting for the study's survey cycle in Table 3 (OR = 1.65, p = 0.017, CI 1.10–2.47).

## Discussion

In this nationally representative sample, individuals aged 18 to 39 years were more likely to have a positive urine CT result if they reported the ED as the place they frequented most often for healthcare. The association between a positive urine CT result and the ED remained significant after controlling for age, gender, race/ethnicity, education, and insurance status. This association has not been previously shown on a population level and may facilitate increased STI screening and diagnosis among this young adult population. This study also highlights ethnic and educational differences noting individuals aged 18 to 39 years were more likely to have a positive urine CT result if they were non-Hispanic Black or Mexican American or had an education level of high school graduate (GED equivalent) or less.

Our findings suggest that additional investigation is warranted to examine the relationship between individuals at risk for STIs and the healthcare settings they frequent. The place individuals visit most often for healthcare (i.e. continuity of care) is known to impact their health outcomes including STI screening and treatment [14]. In our study, young adults frequenting the ED had a 4.3% prevalence of CT. These findings are consistent with a single site study of young adults aged 18–30 years presenting to an urban ED for evaluation of non-genitourinary complaints, finding an 8.7% prevalence of CT [15]. In a retrospective study of women aged 19–25 years presenting to an urban ED with dysuria, Wilbanks et al. found that only 20% of

**Table 2. Multivariable analysis of *Chlamydia trachomatis* urine samples among NHANES participants, 1999 to 2016 (n = 19,275; N = 85,836,321).**

| Variable | Bivariate Analysis: OR (95% CI) | p | Multivariable Analysis: OR (95% CI) | p |
|---|---|---|---|---|
| **Place of Healthcare** | | | | |
| Doctor's Office/Clinic | Reference | | Reference | |
| ED | **2.68 (1.81 to 3.95)** | 0.000 | **1.58 (1.07 to 2.36)** | 0.022 |
| Other | 0.76 (0.27 to 2.10) | 0.591 | 0.84 (0.29 to 2.37) | 0.735 |
| No Place | **1.34 (1.07 to 1.68)** | 0.010 | 1.22 (0.96 to 1.55) | 0.105 |
| **Age** | 0.78 (0.64 to0.95) | 0.014 | 0.85 (0.69 to 1.03) | 0.097 |
| **Age x Age** | 1.00 (1.00 to 1.01) | 0.131 | 1.00 (0.99 to 1.01) | 0.385 |
| **Sex** | | | | |
| Male | Reference | | Reference | |
| Female | 1.11 (0.90 to 1.38) | 0.326 | 1.18 (0.93 to 1.51) | 0.169 |
| **Race/Ethnicity** | | | | |
| Non-Hispanic White | Reference | | Reference | |
| Non-Hispanic Black | **5.18 (3.86 to 6.96)** | 0.000 | **4.38 (3.20 to 6.00)** | 0.000 |
| Mexican American | **2.36 (1.67 to 3.33)** | 0.000 | **1.83 (1.22 to 2.74)** | 0.004 |
| Other Hispanic | **1.79 (1.13 to 2.86)** | 0.014 | 1.59 (0.97 to 2.59) | 0.064 |
| Other Race/Ethnicity | 1.47 (0.87 to 2.49) | 0.146 | 1.54 (0.92 to 2.60) | 0.102 |
| **Education Level** | | | | |
| College Grad or higher | Reference | | Reference | |
| Less than 9th Grade | **3.52 (1.84 to 6.72)** | 0.000 | **2.31 (1.11 to 4.83)** | 0.025 |
| 9th-11th Grade | **4.59 (2.99 to 7.04)** | 0.000 | **2.11 (1.31 to 3.41)** | 0.003 |
| HS Graduate/GED | **3.67 (2.37 to 5.67)** | 0.000 | **2.10 (1.30 to 3.39)** | 0.003 |
| Some College/Assoc. | **2.29 (1.48 to 3.54)** | 0.000 | 1.38 (0.87 to 2.17) | 0.168 |
| **Insurance Status** | | | | |
| Private | Reference | | Reference | |
| Medicaid | **2.63 (1.96 to 3.52)** | 0.000 | 1.23 (0.90 to 1.70) | 0.197 |
| Uninsured | **1.58 (1.24 to 2.02)** | 0.000 | 1.00 (0.76 to 1.32) | 0.991 |

*Age squared allows the model to adjust for the effect of differing ages, which has a non-linear relationship with the independent variable (urine CT).

these women were tested for CT but 21% of them tested positive [16]. Our study builds on the findings that young adults who are at risk for CT are seeking care in the ED and represent a clinically relevant sample of symptomatic and asymptomatic CT infections.

Understanding where young adults obtain sexual healthcare is an important step in addressing rising STI rates. The ED may be a crucial healthcare site to combat the record high rates of STIs as young adults make up the majority of STI cases and are known to frequent the ED for healthcare. However, the ED setting may not be designed to screen large volumes of patients for STIs in comparison to other healthcare settings [1, 6, 17, 18]. In busy ED settings, performing highly sensitive nucleic acid amplification tests (NAAT) for CT *that* can be performed on a self-administered vaginal swab or first-void urine samples may increase identification of CT in young adults [19]. As the ED may be the only point of care for several young adults, screenings have the potential to reduce STI rates and prevent the spread to future sexual partners.

Our findings should be considered in the context of certain limitations. This study is of a pooled cross-sectional study and cannot be used to determine causation or directionality. As a self-reported questionnaire, participant answers may be subject to recall bias. Participants in our sample were asked about their usual source of healthcare, however this may not be the same as where they would go to obtain STI screening and treatment. NHANES only collected

**Table 3. Multivariable analysis of *Chlamydia trachomatis* urine samples among NHANES participants with survey cycle, 1999 to 2016 (n = 19,275; N = 85,836,321).**

| Variable | Multivariable Analysis: OR (95% CI) | *p* |
|---|---|---|
| **Place of Healthcare** | | |
| Doctor's Office/Clinic | Reference | |
| ED | **1.65 (1.10 to 2.47)** | 0.017 |
| Other | 0.84 (0.30 to 2.41) | 0.075 |
| No Place | 1.22 (0.96 to 1.55) | 0.100 |
| **Survey Cycle** | | |
| 1999–2000 | Reference | |
| 2001–2002 | 0.62 (0.36 to 1.07) | 0.087 |
| 2003–2004 | 0.76 (0.47 to 1.22) | 0.254 |
| 2005–2006 | 0.51 (0.30 to 0.86) | 0.013 |
| 2007–2008 | 0.57 (0.36 to 0.92) | 0.023 |
| 2009–2010 | 0.60 (0.37 to 0.95) | 0.029 |
| 2011–2012 | 0.73 (0.50 to 1.05) | 0.092 |
| 2013–2014 | 0.66 (0.41 to 1.05) | 0.079 |
| 2014–2016 | 0.56 (0.36 to 0.95) | 0.031 |
| **Age** | 0.85 (0.70 to 1.03) | 0.102 |
| **Age x Age** | 1.00 (1.00 to 1.01) | 0.404 |
| **Sex** | | |
| Male | Reference | |
| Female | 1.18 (0.93 to 1.50) | 0.175 |
| **Race/Ethnicity** | | |
| Non-Hispanic White | Reference | |
| Non-Hispanic Black | 4.45 (3.25 to 6.09) | 0.000 |
| Mexican American | 1.87 (1.26 to 2.78) | 0.002 |
| Other Hispanic | 1.55 (0.94 to 2.54) | 0.085 |
| Other Race | 1.54 (0.91 to 2.62) | 0.106 |
| **Education Level** | | |
| College Grad or higher | Reference | |
| Less than 9th Grade | 2.29 (1.08 to 4.83) | 0.030 |
| 9th-11th Grade | 2.04 (1.26 to 3.30) | 0.004 |
| HS Graduate/GED | 2.07 (1.28 to 3.36) | 0.003 |
| Some College/Assoc. | 1.36 (0.86 to 2.15) | 0.194 |
| **Insurance Status** | | |
| Private | Reference | |
| Medicaid | 1.24 (0.90 to 1.70) | 0.197 |
| Uninsured | 1.00 (0.76 to 1.32) | 0.991 |

*Age squared allows the model to adjust for the effect of differing ages, which has a non-linear relationship with the independent variable (urine CT).

urine samples from individuals aged 18 to 39 years (available in public data), therefore this study may not be generalizable to other age groups. Only urine CT samples were collected by NHANES, therefore this study may not be generalizable to all sites of CT infections, such as oral. NHANES participants are in the US and therefore may not be generalizable to other countries or healthcare models. The data from the nine most recent NHANES cohorts (1999 to 2016) used in this study encompass the pre- and post-enactment of the Affordable Care Act that largely took effect January 1, 2014.

Young adults make up the majority of STI cases and are known to frequent the ED for healthcare [1, 17, 20]. These cases represent a sizeable number of patients with CT that could go undiagnosed without self-administered screening in the ED. Additional research on cost-effectiveness of ED screening and improving primary care options for young adults is warranted. As STI rates continue to rise in the US, collaborations between EDs and sexual health researchers have the potential to decrease barriers to STI screening and treatment.

## Author Contributions

**Conceptualization:** Cornelius D. Jamison, Tammy Chang.

**Data curation:** Cornelius D. Jamison, Caroline R. Richardson, Hwajung Choi, Tammy Chang.

**Formal analysis:** Cornelius D. Jamison, Caroline R. Richardson, Hwajung Choi, Tammy Chang.

**Funding acquisition:** Cornelius D. Jamison.

**Investigation:** Cornelius D. Jamison.

**Methodology:** Cornelius D. Jamison, Hwajung Choi, Tammy Chang.

**Resources:** Cornelius D. Jamison, Margaret Greenwood-Ericksen.

**Software:** Cornelius D. Jamison, Caroline R. Richardson, Hwajung Choi.

**Supervision:** Cornelius D. Jamison, Tammy Chang.

**Validation:** Cornelius D. Jamison, Margaret Greenwood-Ericksen, Caroline R. Richardson.

**Visualization:** Cornelius D. Jamison.

**Writing – original draft:** Cornelius D. Jamison, Tammy Chang.

**Writing – review & editing:** Cornelius D. Jamison, Margaret Greenwood-Ericksen, Caroline R. Richardson, Hwajung Choi, Tammy Chang.

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
