## [Decision Letter · Decision Letter 0]

22 Jul 2020

PONE-D-20-16986

Association between Chlamydia and Routine Place for Healthcare in the United States: NHANES 1999-2016

PLOS ONE

Dear Dr. Jamison,

Thank you for submitting your manuscript to PLOS ONE. After careful consideration, we feel that it has merit but does not fully meet PLOS ONE’s publication criteria as it currently stands. Therefore, we invite you to submit a revised version of the manuscript that addresses the points raised during the review process.

In addition to the comments of the reviewers, with which I agree, I have also added a few comments to inprove the current manuscript.

We look forward to receiving your revised manuscript.

Kind regards,

Sylvia Maria Bruisten, Ph.D

Academic Editor

PLOS ONE

Journal Requirements:

2. IN your Methods section, please include more information on the Logistic regression details to ensure the reproducibility of your analysis.

3.In your Data Availability statement, you have not specified where the minimal data set underlying the results described in your manuscript can be found. PLOS defines a study's minimal data set as the underlying data used to reach the conclusions drawn in the manuscript and any additional data required to replicate the reported study findings in their entirety. All PLOS journals require that the minimal data set be made fully available. For more information about our data policy, please see http://journals.plos.org/plosone/s/data-availability.

Additional Editor Comments (if provided):

In addition to the comments of both reviewers, with which I agree, there are a few more comments to make.

In Table 1 the mean age is given. I suggest to give the median age and range. Also I suggest to change the term 'race' into 'ethnicity' (same goes for Table 2 and in the text).

Also it would be informative to add the Chlamydia trachomatis prevalences in Table 1 for each group.

Reviewers' comments:

Reviewer's Responses to Questions

**Comments to the Author**

1. Is the manuscript technically sound, and do the data support the conclusions?

Reviewer #1: Partly

Reviewer #2: Yes

2. Has the statistical analysis been performed appropriately and rigorously? 

Reviewer #1: Yes

Reviewer #2: No

3. Have the authors made all data underlying the findings in their manuscript fully available?

Reviewer #1: No

Reviewer #2: No

4. Is the manuscript presented in an intelligible fashion and written in standard English?

Reviewer #1: No

Reviewer #2: Yes

5. Review Comments to the Author

Reviewer #1: This manuscript evaluates the association between a positive C. trachomatis (CT) screening in individuals aged 18-38 year and self-reporting of their usual place to seek healthcare. They found that 3.3% of included individuals reported visiting the emergency department for healthcare was associated with having a positive urine CT screening result in multivariate logistic regression. This data might help to inform more STI screening approaches at emergency department to help reduce the number of CT infections. Nevertheless, there are some points that need to be clarified and adjusted to improve this manuscript. The authors should especially elaborate on the importance of the association between having a positive CT screen and visiting the ED. Especially given that 96.7% of individuals apparently seek healthcare elsewhere and that 91.5% of the detected CT positive cases seek healthcare elsewhere suggesting that emergency departments do not see the majority of CT cases.

Abstract

1. Page 2, line 55: The abbreviation ‘ED’ should be explained and written in full before first use.

2. Page 2, line 60: Can the authors elaborate on how the association of a positive CT screening result with and ED visit can help improve STI clinical and policy interventions? Especially given that 96.7% of individuals apparently seek healthcare elsewhere and that 91.5% of the detected CT positive cases seek healthcare elsewhere suggesting that emergency departments do not see the majority of CT cases.

Introduction

3. Page 3, line 15: The abbreviation ‘ED’ should be explained and written in full before first use in the introduction as well.

4. Page 3, line 19: What kind of diagnostic CT screening assay was used?

Methods

5. Page 4, line 35: The authors should clarify what is meant with ‘no place of care’. How can persons undergo medical examination, CT testing and treatment without visiting a place of care?

6. Page 4, line 31. The question ‘what place do you go to most often’ refers to the place visited in general for any kind of healthcare. This question was not included specifically for STI tests. Could participants have answered differently to which place they visited most often if they were asked this question specifically in the context of receiving an STI screening and could results therefore be biased? The authors should discuss this in the discussion.

7. Page 4, line 43: There should be a brief description of the assays used to perform urine CT screening. It’s unclear if these were all nucleic amplification tests or other diagnostic assays. Also, reference 9 refers to an overall CDC website but the link does not refer to the actual report. The authors should include a link to the actual report that is being referenced to.

Results:

8. Page 5, line 57: How many individuals participated in total? Only the number of individuals with complete medical examination data was included which suggest everyone with incomplete data was excluded? This is not mentioned in the Methods. Authors should include the total number of individuals and the number of excluded participants. Also, if authors only include individuals with complete medical examination data, how can they have missing covariates?

9. Page 5, line 64: It’s hard to understand why 25% of the people usually going to ‘no place’. Does this mean that a quarter of the population included does not get healthcare? Can the authors please clarify what is meant with ’no place of care’.

10. Table 1: Suggest including a variable ‘Positive CT screen’ to show what percentage of individuals tested CT positive at each place of care and a variable to indicate the total number of individuals that reported visiting each healthcare site. Also, please check this table. The overall percentages for education do not sum up to 100%.

11. Page 6, Line 75: Please remove reference to Table 2. An OR=2.68 has not been reported in Table 2. I think only the adjusted OR is mentioned in Table 2?

12. Table 2: Suggest including the bivariate analysis, unadjusted and adjusted results in this table to help clarify what ORs are actually shown in Table 2.

Discussion:

13. Page 8, line 100. The percentage of cases at each site should be mentioned in results first before it’s being discussed.

14. Only 3.3% of individuals reported going to the ED constituting only 8.5% of the total number of positive CT cases. Could it be that a bigger proportion of positive CT cases are found on the ED as people visiting the ED might wait longer to access healthcare and often do so with more serious problems or in a later stage of disease where it cannot longer be ignored, whereas people visiting other healthcare facilities may do so on a more routine basis and in an earlier stage or for preventive reasons? Could this difference in healthcare visiting behaviour have biased the association with ED compared to other places of care?

15. Page 9, line 117-125. It should be made clear that these limitations cannot be generalised to other countries as they only reflect the healthcare situation in the US.

Reviewer #2: Thank you for giving me the opportunity to review this interesting work. The authors looked at determinants of chlamydia infection in NHANES data. They found that respondents using the ED as a usual place for healthcare have higher odds of a chlamydia infection compared to people visiting the doctor’s office. This makes the ED a potential venue for increased routine chlamydia screening.

I have some comments on the methods used:

1) Why was survey cycle (time) not included as a (continuous) variable in the analyses? As urine samples were processed using different assays throughout the years, and years covering the pre- and post-enactment of the Affordable Care Act were used, inclusion of survey cycle seems important.

2) Table 1 clearly shows that people who visit the ED are more often Non-Hispanic Black, lower educated and uninsured. I suggest the authors to test this statistically. Furthermore, this association should also be mentioned and explained in the discussion section. Furthermore, the authors should test for interaction and correct for it if needed in the multivariate analyses.

Other comments:

3) Methods: It is unclear how question HUQ030 is used to determine to place of healthcare. Is this used in combination with HUQ040? And if so, how?

4) Methods, lines 32-33: be explicit where “the question” refers to

5) Methods: Please provide the exact question HUQ041

6) In the methods section a bivariate analyses is described, but no results are shown. Please add this.

7) Table 1: please explain the meaning of (Ref), as there is no statistical analyses performed.

8) Please add more explanation in the methods section for the addition of the age X age variable in Table 2.

9) Table 2: please provide a more informative title.

6. PLOS authors have the option to publish the peer review history of their article (what does this mean?). If published, this will include your full peer review and any attached files.

Reviewer #1: No

Reviewer #2: No

---

## [Author Response · Author response to Decision Letter 0]

2 Nov 2020

September 14, 2020

PLOS ONE

Joerg Heber, PhD

Editor-in-Chief

RE: Manuscript Number PONE-D-20-16986

Association between Chlamydia and Routine Place for Healthcare in the United States: NHANES 1999-2016

PLOS ONE

Dear Dr. Joerg Heber,

Thank you for your email dated July 22, 2020 regarding our manuscript “Association between Chlamydia and Routine Place for Healthcare in the United States: NHANES 1999-2016”. We greatly appreciate the suggestions and have edited our manuscript accordingly. Our response to the comments provided are detailed below. Please accept our revised manuscript for continued consideration for inclusion as a Research Article in PLOS ONE.

We appreciate your continued consideration of our work for publication in PLOS ONE.

Sincerely, 

Cornelius D. Jamison, MD, MSPH, MS

Lecturer

Department of Family Medicine

University of Michigan

jcorneli@med.umich.edu

Additional Editor Comments (if provided):

In addition to the comments of both reviewers, with which I agree, there are a few more comments to make.

1. In Table 1 the mean age is given. I suggest to give the median age and range. 

RESPONSE:

Table 1 now includes median age (median age = 28.6). The age range is included in the Methods (age 18-39). 

2. Also, I suggest to change the term 'race' into 'ethnicity' (same goes for Table 2 and in the text).

RESPONSE: 

Thank you for this comment. We have made changes throughout the tables and text to “race/ethnicity” to remain consistent with NHANES labeling of the variable. 

3. Also it would be informative to add the Chlamydia trachomatis prevalences in Table 1 for each group.

RESPONSE:

We agree that this information would be valuable and have changed the table to reflect this information with Chlamydia trachomatis (CT) prevalence. 

Reviewers' Comments:

Reviewer #1: 

This manuscript evaluates the association between a positive C. trachomatis (CT) screening in individuals aged 18-38 year and self-reporting of their usual place to seek healthcare. They found that 3.3% of included individuals reported visiting the emergency department for healthcare was associated with having a positive urine CT screening result in multivariate logistic regression. This data might help to inform more STI screening approaches at emergency department to help reduce the number of CT infections. Nevertheless, there are some points that need to be clarified and adjusted to improve this manuscript. The authors should especially elaborate on the importance of the association between having a positive CT screen and visiting the ED. Especially given that 96.7% of individuals apparently seek healthcare elsewhere and that 91.5% of the detected CT positive cases seek healthcare elsewhere suggesting that emergency departments do not see the majority of CT cases.

Abstract

1. Page 2, line 55: The abbreviation ‘ED’ should be explained and written in full before first use.

RESPONSE:

Thank you for this comment. Emergency Department has been written in full before first use.

2. Page 2, line 60: Can the authors elaborate on how the association of a positive CT screening result with and ED visit can help improve STI clinical and policy interventions? Especially given that 96.7% of individuals apparently seek healthcare elsewhere and that 91.5% of the detected CT positive cases seek healthcare elsewhere suggesting that emergency departments do not see the majority of CT cases.

RESPONSE:

Thank you for this comment. We agree that our study focuses on a small sample of individuals. However, our findings show that in a nationally representative dataset, individuals found to have an incidental CT infection are more likely to report the ED as their usual place of healthcare, even after controlling for other variables. This is important because individuals that visit the ED for their usual care are less likely to receive regular STI screenings. The majority of individuals with CT infections are asymptomatic resulting in missed cases that lead to further spread of the infection. This study is important because those who go to the ED for usual care are not likely to get screened for STIs unless that is their chief complaint. Therefore, many individuals with STIs could be missed leading to increased spread unless EDs consider the simple and inexpensive practice of screening.

We have highlighted the importance of this association in the Abstract. The statement on Page 2, Line 59-63 now reads: 

“Individuals having a positive urine CT screen are associated with using the ED as a usual source for healthcare. Understanding this association has the potential to improve STI clinical and policy interventions as the ED may be a critical site in combatting the record high rates of STIs.”

Introduction

3. Page 3, line 15: The abbreviation ‘ED’ should be explained and written in full before first use in the introduction as well.

RESPONSE:

Thank you for this comment. Emergency Department has been written in full. 

4. Page 3, line 19: What kind of diagnostic CT screening assay was used?

RESPONSE:

Thank you for this comment. We have updated the Methods section to reflect the different assays NHANES used as the waves progressed. The statement on Page 4, Line 46-49 that now reads: 

“NHANES waves 1999-2002 used the LCx Chlamydia trachomatis assay, 2003-08 used the BDProbeTec CT Chlamydia trachomatis and Neisseria gonorrhoeae Amplified DNA Assays, waves 2009-16 used the Gen-Probe APTIMA Combo 2 Chlamydia trachomatis Assay.”

Methods

5. Page 4, line 35: The authors should clarify what is meant with ‘no place of care’. How can persons undergo medical examination, CT testing and treatment without visiting a place of care?

RESPONSE: 

To better clarify the methods of our study, the National Health and Nutrition Examination Survey (NHANES) combines interviews and physical examinations with laboratory testing to assess the health and nutritional status of adults and children in the United States. These participants do not receive medical care during this survey. In our case, the participants provided urine samples that were tested for CT. The participants testing positive for CT, may have been unaware of their infection and were not seeking care at the time of their NHANES participation. ‘No place of care’ means the participants reported no usual place of care. To better clarify the Methods section, we have added information on Page 5, Line 31 that now reads:

“NHANES participants were not seeking nor being provided medical care at the time of their participation.”

6. Page 4, line 31. The question ‘what place do you go to most often’ refers to the place visited in general for any kind of healthcare. This question was not included specifically for STI tests. Could participants have answered differently to which place they visited most often if they were asked this question specifically in the context of receiving an STI screening and could results therefore be biased? The authors should discuss this in the discussion.

RESPONSE:

We understand and appreciate this concern. NHANES does not specifically ask participants which place they visit most often in the context of obtaining STI screening. We have added the following statement to our Discussion section. The statement on Page 11, Line 154 now reads: 

“Participants in our sample were asked about their usual source of healthcare, however this may not be the same as where they would go to obtain STI screening and treatment.”

7. Page 4, line 43: There should be a brief description of the assays used to perform urine CT screening. It’s unclear if these were all nucleic amplification tests or other diagnostic assays. Also, reference 9 refers to an overall CDC website but the link does not refer to the actual report. The authors should include a link to the actual report that is being referenced to.

RESPONSE:

Thank you for the feedback. NHANES used different assays to perform urine CT screening as the waves progressed. We have updated the references to include NHANES’ laboratory procedure manual the urine CT method. We have added a statement on Page 4, Line 46-49 that now reads: 

“NHANES waves 1999-2002 used the LCx Chlamydia trachomatis assay, 2003-08 used the BDProbeTec CT Chlamydia trachomatis and Neisseria gonorrhoeae Amplified DNA Assays, waves 2009-16 used the Gen-Probe APTIMA Combo 2 Chlamydia trachomatis Assay.”

Results:

8. Page 5, line 57: How many individuals participated in total? Only the number of individuals with complete medical examination data was included which suggest everyone with incomplete data was excluded? This is not mentioned in the Methods. Authors should include the total number of individuals and the number of excluded participants. Also, if authors only include individuals with complete medical examination data, how can they have missing covariates?

RESPONSE:

To better clarify the methods and results of our study, the National Health and Nutrition Examination Survey (NHANES) is a complex study design that combines interviews and physical examinations with laboratory testing to assess the health and nutritional status of adults and children in the United States. The complex study design of NHANES uses a subset of the US population, representing the unweighted number of individuals. The weighting measurements used in analysis account for the US population. This has been added to our Methods section on Page 5, Line xxx. 

The total number of NHANES participants in the 9 waves (1999-2016) was 92,062 of which 20,978 participants were 18-39 years of age. The total number of participants in the 9 waves providing urine samples for analysis of Chlamydia trachomatis was 19,514. During multivariable analysis, 239 individual observations had missing covariate data, resulting in the 19,275 individual observations discussed in the Results section. We have modified the statement on Page 5, Line 57-59 for better clarification, that now reads:

“Data collected from NHANES between 1999-2016 resulted in an unweighted sample size of 19,514 individuals (weighted N = 86.8 million) aged 18 to 39 years with complete CT urine sample data.” 

9. Page 5, line 64: It’s hard to understand why 25% of the people usually going to ‘no place’. Does this mean that a quarter of the population included does not get healthcare? Can the authors please clarify what is meant with ’no place of care’.

RESPONSE:

We understand and appreciate this concern. ‘Place of healthcare’ categories were developed based on NHANES participant’s responses to HUQ040 and HUQ041. We determined that the participants who responded to NHANES questionnaire stating they did not go to one place most often for healthcare or refused to answer or did not know where they went for healthcare would be categorized as ‘no place.’ This does not imply that these participants do not receive healthcare, however based on their response we determined they do not identify a specific place they go to get their healthcare. Additional detail to the development of the ‘no place’ category can be found under the response to Reviewer #1’s comment #5. 

10. Table 1: Suggest including a variable ‘Positive CT screen’ to show what percentage of individuals tested CT positive at each place of care and a variable to indicate the total number of individuals that reported visiting each healthcare site. Also, please check this table. The overall percentages for education do not sum up to 100%.

RESPONSE:

We agree that this information would be valuable and have changed the table to reflect this information with Chlamydia trachomatis prevalence. 

Thank you for this comment, we have corrected the overall education percentages. 

11. Page 6, Line 75: Please remove reference to Table 2. An OR=2.68 has not been reported in Table 2. I think only the adjusted OR is mentioned in Table 2?

RESPONSE:

Thank you for this comment, we have added the bivariate analysis to Table 2. 

12. Table 2: Suggest including the bivariate analysis, unadjusted and adjusted results in this table to help clarify what ORs are actually shown in Table 2.

RESPONSE:

Thank you for this comment, we have added the bivariate analysis to Table 2.

Discussion:

13. Page 8, line 100. The percentage of cases at each site should be mentioned in results first before it’s being discussed.

RESPONSE:

We understand and appreciate this concern. We have revised this statement on Page 10, Line 132 that now reads: 

“In our study, young adults frequenting the ED had a 4.3% prevalence of CT.”

14. Only 3.3% of individuals reported going to the ED constituting only 8.5% of the total number of positive CT cases. Could it be that a bigger proportion of positive CT cases are found on the ED as people visiting the ED might wait longer to access healthcare and often do so with more serious problems or in a later stage of disease where it cannot longer be ignored, whereas people visiting other healthcare facilities may do so on a more routine basis and in an earlier stage or for preventive reasons? Could this difference in healthcare visiting behaviour have biased the association with ED compared to other places of care?

RESPONSE:

We greatly appreciate these concerns, as they are important to understanding social determinants of health. However, the NHANES combines interviews and physical examinations with laboratory testing to assess the health and nutritional status of adults and children in the United States. To better clarify the methods of our study, these participants do not receive medical care and are being cared for at specific places of healthcare during this survey.

15. Page 9, line 117-125. It should be made clear that these limitations cannot be 

generalised to other countries as they only reflect the healthcare situation in the US.

RESPONSE:

Thank you for this comment, we have added a statement to Discussion section on Page 11, Line 159 that now reads: 

“NHANES participants are in the US and therefore may not be generalizable to other countries or healthcare models.” 

Reviewer #2: 

Thank you for giving me the opportunity to review this interesting work. The authors looked at determinants of chlamydia infection in NHANES data. They found that respondents using the ED as a usual place for healthcare have higher odds of a chlamydia infection compared to people visiting the doctor’s office. This makes the ED a potential venue for increased routine chlamydia screening.

I have some comments on the methods used:

1) Why was survey cycle (time) not included as a (continuous) variable in the analyses? As urine samples were processed using different assays throughout the years, and years covering the pre- and post-enactment of the Affordable Care Act were used, inclusion of survey cycle seems important.

RESPONSE:

Thank you for this comment. We conducted additional multivariate analysis models that control for NHANES’ survey cycle to see if there was any survey specific effect (e.g., different assays across surveys) that influenced the main relationship (association between healthcare place and CT). Results of these additional models were consistent with results without survey cycle control with similar effect size. 

2) Table 1 clearly shows that people who visit the ED are more often Non-Hispanic Black, lower educated and uninsured. I suggest the authors to test this statistically. Furthermore, this association should also be mentioned and explained in the discussion section. Furthermore, the authors should test for interaction and correct for it if needed in the multivariate analyses.

RESPONSE:

We understand and appreciate this comment. This information is valuable to the discussion. We performed bivariate analysis and have added these results to Table 2. We also conducted different multivariate analysis models including a multinomial logistic regression to better understand these covariate interactions. Results of these additional models remained statistically significant and did not alter our findings. We have included more discussion regarding these findings in the Discussion section, Page 10, Line 125 now reads: 

“This study also highlights ethnic and educational differences noting individuals aged 18 to 39 years were more likely to have a positive urine CT result if they were non-Hispanic Black or Mexican American or had an education level of high school graduate (GED equivalent) or lower.”

Other comments:

3) Methods: It is unclear how question HUQ030 is used to determine to place of healthcare. Is this used in combination with HUQ040? And if so, how?

RESPONSE:

Thank you for this comment. ‘Place of healthcare’ categories were developed based on NHANES participant’s responses to HUQ040/HUQ041. For ‘doctor’s office/clinic’ we combined clinic or health center, doctor’s office or HMO, and hospital outpatient department. For ‘ED’ we used hospital emergency room, for ‘other’ we used some other place, and for ‘no place’ we combined doesn’t go to one place most often, refused and don’t know. We decided that the participants who responded to NHANES questionnaire stating they did not go to one place most often for healthcare or refused to answer or did not know would be categorized as ‘no place.’ Additional detail to the development of the ‘no place’ category can be found under the response to Reviewer #1’s comment #5 and #9. 

4) Methods, lines 32-33: be explicit where “the question” refers to

RESPONSE:

We appreciate the comment and reworded the statement for clarity. The statement on Page 4, Line 33-35 now reads: 

“Beginning in 2013 the question for variable HUQ040 was modified but collected the same information (variable HUQ041).”

5) Methods: Please provide the exact question HUQ041

RESPONSE:

We appreciate the comment and have added the exact question to the Methods section for clarity. The statement on Page 4, Line 34 now reads:

If the participants answered “Yes” to HUQ030, then they were asked "What kind of place is it - a clinic, doctor's office, emergency room, or some other place?" If they provided any answer other than “Yes” to HUQ030, then they were asked "What kind of place do you go to most often - a clinic, doctor's office, emergency room, or some other place?"

6) In the methods section a bivariate analyses is described, but no results are shown. Please add this.

RESPONSE:

Thank you for this comment, we have added the bivariate analysis to Table 2.

7) Table 1: please explain the meaning of (Ref), as there is no statistical analyses performed.

RESPONSE:

Thank you for this comment, this was a mistake and has been removed from the table. 

8) Please add more explanation in the methods section for the addition of the age X age variable in Table 2.

RESPONSE:

We included the variable age squared as this is one of the standard ways to address a potential non-linear effect of age. Adding the age squared variable allows us to model the effect of differing ages. We clarified this in the Methods section. The statement on Page 6, Line 67 now reads: 

“To address the potential non-linear effect of age, we included the quadratic form age squared (shown as ‘Age x Age’ in Table 2) rather than linear form, age alone.”

9) Table 2: please provide a more informative title.

RESPONSE:

We have modified the title which now reads: 

“Table 2. Multivariable Analysis of Chlamydia trachomatis urine samples among NHANES participants, 1999 to 2016 (n = 19,275; N = 85,836,321).”

---

## [Decision Letter · Decision Letter 1]

25 Nov 2020

PONE-D-20-16986R1

Association between Chlamydia and Routine Place for Healthcare in the United States: NHANES 1999-2016

PLOS ONE

Dear Dr. Jamison,

Thank you for submitting your manuscript to PLOS ONE. After careful consideration, we feel that it has merit but does not fully meet PLOS ONE’s publication criteria as it currently stands. Therefore, we invite you to submit a revised version of the manuscript that addresses the points raised during the review process.

One of the reviewers correctly states that major comments were not adequately addressed. Please add the requested data to the manuscript.

The other reviwer is satisfied with your answers and actions. 

We look forward to receiving your revised manuscript.

Kind regards,

Sylvia Maria Bruisten, Ph.D

Academic Editor

PLOS ONE

Reviewers' comments:

Reviewer's Responses to Questions

**Comments to the Author**

1. If the authors have adequately addressed your comments raised in a previous round of review and you feel that this manuscript is now acceptable for publication, you may indicate that here to bypass the “Comments to the Author” section, enter your conflict of interest statement in the “Confidential to Editor” section, and submit your "Accept" recommendation.

Reviewer #2: (No Response)

Reviewer #3: All comments have been addressed

2. Is the manuscript technically sound, and do the data support the conclusions?

Reviewer #2: Partly

Reviewer #3: Yes

3. Has the statistical analysis been performed appropriately and rigorously? 

Reviewer #2: No

Reviewer #3: Yes

4. Have the authors made all data underlying the findings in their manuscript fully available?

Reviewer #2: Yes

Reviewer #3: Yes

5. Is the manuscript presented in an intelligible fashion and written in standard English?

Reviewer #2: Yes

Reviewer #3: Yes

6. Review Comments to the Author

Reviewer #2: Thank you for giving me the opportunity to see a revised version of the manuscript. I appreciate the time and effort the authors put into the revisions. However, I feel that both my major comments were not addressed adequately.

My first major comment was about survey cycle not being included in the analyses. Although the authors did address this issue in their response (by saying it did not change the results), they did not change anything to the manuscript. I therefore encourage the authors to either included survey cycle in the main analyses in the manuscript, or as a sensitivity analyses in a supplement.

My second major comment addressed possible interaction between covariates in the model (which you would expect by looking at Table 1). Performing a bivariate analyses between a covariate and CT does not address this issue. Furthermore, the authors write in their response something about “additional” models, but again, no results are shown or details are given.

Reviewer #3: I am satisfied with the authors corrections in this revised version of the manuscript and have no further comments.

7. PLOS authors have the option to publish the peer review history of their article (what does this mean?). If published, this will include your full peer review and any attached files.

Reviewer #2: No

Reviewer #3: No

---

## [Author Response · Author response to Decision Letter 1]

16 Apr 2021

Dear Drs. Chenette & Bruisten,

Thank you for your email dated November 25, 2020 regarding our manuscript “Association between Chlamydia and Routine Place for Healthcare in the United States: NHANES 1999-2016”. We greatly appreciate the suggestions and have edited our revised manuscript accordingly. Our response to the comments provided are detailed below. Please accept our revised manuscript for continued consideration for inclusion as a Research Article in PLOS ONE.

We appreciate your continued consideration of our work for publication in PLOS ONE.

Sincerely, 

Cornelius D. Jamison, MD, MSPH, MS

Assistant Professor

Department of Family Medicine

University of Michigan

jcorneli@med.umich.edu

---

## [Editor Report · Decision Letter 2]

21 Apr 2021

Association between Chlamydia and Routine Place for Healthcare in the United States: NHANES 1999-2016

PONE-D-20-16986R2

Dear Dr. Jamison,

We’re pleased to inform you that your manuscript has been judged scientifically suitable for publication and will be formally accepted for publication once it meets all outstanding technical requirements.

Kind regards,

Sylvia Maria Bruisten, Ph.D

Academic Editor

PLOS ONE

Additional Editor Comments (optional):

The manuscript has now been revised to almost full satisfaction.

A minor point: in Resultsline 78 firts paragraph: please replace ' mean' by 'median' as this is what is shown in Table 1.

Second comment: the new Table which is now added as Supplementary table: please add this as a regular Table 3 in the results section and also refer to it as such in the text.
---

## [Editor Report · Acceptance letter]

30 Apr 2021

PONE-D-20-16986R2 

Association between Chlamydia and Routine Place for Healthcare in the United States: NHANES 1999-2016 

Dear Dr. Jamison:

I'm pleased to inform you that your manuscript has been deemed suitable for publication in PLOS ONE. Congratulations! Your manuscript is now with our production department. 

Kind regards, 

on behalf of

Dr. Sylvia Maria Bruisten 

Academic Editor

PLOS ONE